# Changes in Metabolomic Profiles Induced by Switching from an Erythropoiesis-Stimulating Agent to a Hypoxia-Inducible Factor Prolyl Hydroxylase Inhibitor in Hemodialysis Patients: A Pilot Study

**DOI:** 10.3390/ijms241612752

**Published:** 2023-08-13

**Authors:** Kimio Watanabe, Emiko Sato, Eikan Mishima, Shinobu Moriya, Takuma Sakabe, Atsuya Sato, Momoko Fujiwara, Takuya Fujimaru, Yugo Ito, Fumika Taki, Masahiko Nagahama, Kenichi Tanaka, Junichiro James Kazama, Masaaki Nakayama

**Affiliations:** 1Division of Nephrology and Hypertension, Fukushima Medical University, Fukushima 960-1295, Japan; slopewallboilbear@gmail.com (T.S.); atyuatyu@fmu.ac.jp (A.S.); f-momoko@fmu.ac.jp (M.F.); kennichi@fmu.ac.jp (K.T.); jjkaz@fmu.ac.jp (J.J.K.); 2Kidney Center, St Luke’s International Hospital, Tokyo 104-8560, Japan; fujitaku@luke.ac.jp (T.F.); yito@luke.ac.jp (Y.I.); ftaki@luke.ac.jp (F.T.); managaha@luke.ac.jp (M.N.); nakayama@luke.ac.jp (M.N.); 3Division of Clinical Pharmacology and Therapeutics, Faculty of Pharmaceutical Sciences, Graduate School of Pharmaceutical Sciences, Tohoku University, Sendai 980-8578, Japan; emiko.sato.b8@tohoku.ac.jp; 4Division of Nephrology, Rheumatology and Endocrinology, Graduate School of Medicine, Tohoku University, Sendai 980-8575, Japan; eikan@med.tohoku.ac.jp; 5Institute of Metabolism and Cell Death, Helmholtz Zentrum München, 85764 Neuherberg, Germany; 6Clinical Engineering Center, St Luke’s International Hospital, Tokyo 104-8560, Japan; shinora@luke.ac.jp

**Keywords:** metabolomic analysis, erythropoiesis-stimulating agents, hypoxia-inducible factor prolyl hydroxylase inhibitors, pleiotropic effects, hemodialysis, clinical trial

## Abstract

Hypoxia-inducible factor prolyl hydroxylase inhibitors (HIF-PHIs) are a new class of medications for managing renal anemia in patients with chronic kidney disease (CKD). In addition to their erythropoietic activity, HIF-PHIs exhibit multifaceted effects on iron and glucose metabolism, mitochondrial metabolism, and angiogenesis through the regulation of a wide range of HIF-responsive gene expressions. However, the systemic biological effects of HIF-PHIs in CKD patients have not been fully explored. In this prospective, single-center study, we comprehensively investigated changes in plasma metabolomic profiles following the switch from an erythropoiesis-stimulating agent (ESA) to an HIF-PHI, daprodustat, in 10 maintenance hemodialysis patients. Plasma metabolites were measured before and three months after the switch from an ESA to an HIF-PHI. Among 106 individual markers detected in plasma, significant changes were found in four compounds (erythrulose, n-butyrylglycine, threonine, and leucine), and notable but non-significant changes were found in another five compounds (inositol, phosphoric acid, lyxose, arabinose, and hydroxylamine). Pathway analysis indicated decreased levels of plasma metabolites, particularly those involved in phosphatidylinositol signaling, ascorbate and aldarate metabolism, and inositol phosphate metabolism. Our results provide detailed insights into the systemic biological effects of HIF-PHIs in hemodialysis patients and are expected to contribute to an evaluation of the potential side effects that may result from long-term use of this class of drugs.

## 1. Introduction

Hypoxia-inducible factor prolyl hydroxylase inhibitors (HIF-PHIs) are a new class of oral medications for patients with anemia resulting from chronic kidney disease (CKD) [1]. HIF-PHIs include several drugs, such as daprodustat, roxadustat, and vadadustat, that have been clinically approved for the treatment of renal anemia in CKD patients; since 2020, the erythropoietic effect and safety of these compounds have been investigated in multiple large-scale Phase 3 randomized clinical trials (RCTs) enrolling CKD patients undergoing hemodialysis (HD), peritoneal dialysis (PD), or no dialysis [2,3,4,5,6,7,8,9,10,11,12,13]. In these trials, the erythropoietic effect was non-inferior to that seen with erythropoiesis-stimulating agents (ESAs), a conventional class of intravenous (IV) therapeutic agents used for renal anemia in CKD. The incidence of adverse events, including cardiovascular events, with HIF-PHIs was similar to that seen with ESAs.

In addition to its erythropoietic effects, the HIF/prolyl hydroxylase domain oxygen-sensing pathway plays a critical role in regulating biologic processes essential for cell survival, including glucose metabolism, mitochondrial metabolism, angiogenesis, and immune responses [14,15,16,17,18,19]. Thus, activation of the HIF/prolyl hydroxylase domain pathway by HIF-PHIs has multifaceted effects. Indeed, in some of the clinical trials noted above, HIF-PHIs exhibited (in addition to effects on maintaining hemoglobin level) pleiotropic effects of on iron metabolism, such as increases in total iron-binding capacity (TIBC) and the level of transferrin; furthermore, study subjects were reported to show lowered levels of total and low-density lipoprotein (LDL) cholesterol during the trial period of about one year [6,9,13]. However, these diverse effects of HIF-PHIs, both in terms of potential risks and benefits, have not been fully explored in post-market surveillance, as this type of drug has only been available in clinical settings for a short period of time.

In the present study, we explored the systemic effect of HIF-PHIs by conducting a single-center observational study to comprehensively investigate changes in plasma metabolomic profiles in maintenance HD patients switched from an ESA to an HIF-PHI. The results detail the biological effects of HIF-PHIs on the human body, especially in patients with advanced kidney disease. These results also may facilitate evaluation of the potential harms and benefits that can be expected from the long-term use of the HIF-PHI class of medications.

## 2. Results

### 2.1. Patient Disposition and Characteristics

A total of 110 patients on maintenance HD was screened; after exclusions, 15 patients were enrolled in the present trial according to the study protocol, as detailed in the Materials and Methods. Of the 15 enrollees, 10 completed the study (Figure 1). Reasons for study treatment discontinuation include an adverse event (*n* = 1), poor medication adherence (*n* = 3), and loss to follow-up (*n* = 1).

Participants who completed the study (*n* = 10) were those who completed the 3-month treatment after the switch from an ESA to an HIF-PHI. Abbreviations: ESA, erythropoiesis-stimulating agent; HIF-PHI, hypoxia-inducible factor prolyl hydroxylase inhibitor; AE, adverse event.

Baseline characteristics of the 10 HD participants who completed the trial are shown (Table 1). Overall, the participants were all Japanese, with a mean (SD) age of 65.6 (11.3) years and dialysis duration of 5.1 (4.4) years. Hemoglobin level was well controlled (11.1 (1.2) g/dL) with IV or oral iron use in combination with an ESA (darbepoetin alfa) dose of 41.0 (26.8) mcg per week, as confirmed by blood sampling at the beginning of the week. In addition, iron and inflammatory parameters, including ferritin (117.1 (117.6) ng/mL), transferrin saturation (TSAT; 28.5 (10.8)%), and C-reactive protein (CRP; 0.13 (0.17) mg/dL), were adequately controlled among subjects at the study start.

### 2.2. Changes in Parameters Following Switching from an ESA to an HIF-PHI

We conducted paired *t*-tests for the comparison of parameters as assessed before and 3 months after switching from an ESA to an HIF-PHI; these comparisons were performed for each characteristic measured in the 10 HD participants who completed the trial. Among the subjects, nominal (not statistically significant) decreases in systolic blood pressure (BP; *t*(9) = 1.923, *p* = 0.074, r = 0.258) and hemoglobin (Hb; *t*(9) = 1.924, *p* = 0.086, r = −0.249) were confirmed. Six of the 10 who were IV iron users did not need to continue iron therapy by the end of three months of HIF-PHI therapy; the overall fraction of iron users, including those dosed orally or intravenously, decreased from 100% to 40%, without any statistically significant decrease in ferritin or TSAT. Other parameters, such as plasma glucose, LDL cholesterol, triglycerides, and CRP, did not exhibit statistically significant changes compared to baseline. These findings are shown in Table 2.

### 2.3. Overall Metabolomic Analysis of Serum Samples

The levels of plasma metabolites before and 3 months after switching to an HIF-PHI were measured using gas chromatography-mass spectrometry (GC-MS) based the metabolomics approach. An overview of the top 25 enriched metabolite sets that changed in plasma level after switching to an HIF-PHI is shown in Figure 2. This pathway analysis revealed statistically significant changes in multiple parameters, especially among metabolites involved in phosphatidylinositol signaling, ascorbate and aldarate metabolism, and inositol phosphate metabolism. Tryptophan metabolism, as well as pentose and glucuronate interconversions, were also affected.

### 2.4. Specific Compounds Whose Levels Are Potentiated by an HIF-PHI and Are Detected by Comprehensive Metabolomic Analysis

Among 106 individual metabolites detected by non-targeted metabolomic analysis, notable changes were found in nine metabolites following the switch to an HIF-PHI. Details of the changes in each metabolite are shown in Figure 3. Specifically, following the switch to an HIF-PHI, the levels of threonine, N-butyrylglycine, erythrulose, and leucine were increased in a statistically significant fashion. The levels of inositol, phosphoric acid, lyxose, arabinose, and hydroxylamine were nominally decreased; although these effects fell short of statistical significance, these metabolites are notable, given that each participates in a known signaling pathway. A schematic diagram of the crucial pathways altered by a shift from darbepoetin alfa to daprodustat is shown in Figure 4.

Among 106 individual compounds analyzed by comprehensive metabolomic analysis, statistically significant changes were observed in four compounds (threonine, n-butyrylglycine, erythrulose, and leucine). Changes in another five metabolites (inositol, phosphoric acid, lyxose, arabinose, and hydroxylamine) did not reach statistical significance, but those changes each involved a signaling pathway. Comparisons were conducted using two-tailed paired *t*-tests; *p* values of less than 0.05 were considered statistically significant. Abbreviations: ESA, erythropoiesis-stimulating agent; HIF-PHI, hypoxia-inducible factor prolyl hydroxylase inhibitor.

Phosphatidylinositol signaling, ascorbate and aldarate metabolism, and inositol phosphate metabolism were the top hits returned by a pathway analysis of metabolites altered by a shift to an HIF-PHI. Metabolites that were nominally downregulated by the switch to an HIF-PHI are indicated in blue. Abbreviations: I(1)P, 1D-myo-inositol 1-monophosphate; I(3)P, 1D-myo-inositol 3-monophosphate; I(4)P, 1D-myo-inositol 4-monophosphate; I(1,4)P_2_, 1D-myo-inositol. 1,4-biphosphate; I(3,4)P_2_, 1D-myo-inositol. 3,4-biphosphate; I(1,3,4)P_3_, 1D-myo-inositol 1,3,4-trisphosphate; I(1,4,5)P_3_, 1D-myo-inositol 1,4,5-trisphosphate; PI3K, phosphatidylinositol; Akt (synonymous with protein kinase B), a class of kinases originally identified in an AKR mouse thymoma.

## 3. Discussion

In the present study, a comprehensive and non-targeted metabolomic analysis demonstrated that switching from an ESA (darbepoetin alfa) to an HIF-PHI (daprodustat) is associated with changes in the levels of several plasma metabolites. Notably, these altered metabolites included compounds involved in phosphatidylinositol signaling, ascorbate and aldarate metabolism, and inositol phosphate metabolism, resulting in decreases in the accumulation of inositol and glucuronate, in HD patients. Additionally, we confirmed that daprodustat was non-inferior to darbepoetin alfa in its erythropoietic effect in HD patients during the observation period. Our results provide detailed insights into the systemic biological effects of HIF-PHIs in HD patients and are expected to contribute to characterization of the potential side effects that may result from the long-term use of HIF-PHIs.

Our results showed that the plasma level of inositol was nominally decreased following the switch to an HIF-PHI. Inositol is a carbocyclic sugar polyalcohol and plays an important role in many physiological processes, especially those influencing the metabolism of glucose and lipids, as well as metabolism in skeletal muscle [20,21]. For example, inositol mediates cell signal transduction and is involved in 10 important pathways, including galactose metabolism, ascorbate and aldarate metabolism, streptomycin biosynthesis, inositol phosphate metabolism, the biosynthesis of secondary metabolites, microbial metabolism in diverse environments, the biosynthesis of nucleotide sugars, ATP-binding cassette (ABC) transporters, and phosphatidylinositol signaling (https://www.genome.jp/kegg/pathway.html, accessed on 20 July 2023). In fact, decreased levels of inositol have been reported in clinically important conditions such as diabetes, insulin resistance, and metabolic syndrome; indeed, inositol supplementation in animal models has been shown to improve overall physiological performance and to stabilize bone metabolism [22,23,24]. Maintenance of a sufficient inositol level is critical to the resynthesis of the phosphoinositides known to contribute to efficient intracellular signal transduction and the maintenance of such signaling [25,26,27]. The human kidney has been estimated to synthesize 4 g of inositol per day [26]. In combination with our results, these findings suggest that nominal depletion of inositol in HD patients following the shift to an HIF-PHI may mediate the long-term effects of dosing with this class of compounds. These changes are expected to have important effects on glucose, lipid, and skeletal muscle metabolism, as well as on overall physiological performance.

The plasma level of glucuronate also was nominally decreased after switching to daprodustat. Both inositol and glucuronate participate in each of three major signaling pathways identified in the present work, including phosphatidylinositol signaling, ascorbate and aldarate metabolism, and inositol phosphate metabolism, Presumably, the widespread effects of switching to daprodustat reflect changes in the levels of multiple cellular metabolites and not just the depletion of inositol and glucuronate. Regarding phosphatidylinositol signaling, downregulation of this system is thought to contribute to the dysfunction of skeletal muscle and deterioration of kidney function, primarily via the loss of function of phosphoinositide phosphatases [28,29,30]. Less is known about the effects of changes in ascorbate and aldarate metabolism, given that this pathway has been investigated only in animal models. Downregulation of this system is associated with decreases in total antioxidant capacity and increases in the level of malondialdehyde (MDA), a major marker of lipid peroxidation [31]. Regarding inositol phosphate metabolism, this pathway has been shown to affect insulin resistance and sensitivity [32]. This observation may be especially relevant, given that correlations have been reported between CKD and insulin resistance, although the causal nature of this interaction remains unclear [33,34,35].

Four metabolites exhibited statistically significant changes following the switch to an HIF-PHI, including threonine, N-butyrylglycine, erythrulose and leucine. We conjecture that comprehensive metabolomic analysis would indicate that changes in these parameters are associated with relatively small effects on the respective metabolic pathways. However, the effects of these metabolites on renal function, especially in patients with CKD, remain unknown. Further research will be needed to clarify the pathological significance of changes in these metabolites.

In the present study, although the HIF-PHI (daprodustat) was non-inferior compared to the ESA (darbepoetin alfa) in terms of erythropoietic effect during the observation period, the hemoglobin level was nominally lower following the switch. We hypothesize that this effect was due to the prior use of a relatively high dose (exceeding 40 mcg per week) of darbepoetin in the study subjects. In most cases, patients were initiated on a relatively low dose (2 or 4 mg per day) of daprodustat, consistent with the recommendations of the medical package insert. In addition, no obvious effects on iron metabolism, nor abnormalities in lipids, were observed following the switch, consistent with the results of previous RCTs of HIF-PHIs [2,3,4,5,6,7,8,9,10,11,12,13]. None of the subjects who originally required IV iron continued its use after the switch to daprodustat, and ferritin and TSAT were maintained in the study participants; these results presumably reflect an improved efficiency of iron utilization following the switch.

This study also has the aspects of a pilot study, which include clarifying the clinical significance of an HIF-PHI, which was detected by exploratory metabolomic analysis; the calculation of an estimated sample size of this point; feasibility assessment based on patients’ selection and dropout rate; and consideration for the possibility of protocol modification. It was calculated that a minimum sample size of 27 participants was required to clarify significant decrease in inositol level that affected changes in three pathways (phosphatidylinositol signaling, ascorbate and aldarate metabolism, and inositol phosphate metabolism) [36,37]. Here, we applied a power (1–beta error probability) of 0.8, the alpha error probability was 0.05. For the used data, the mean of difference = 0.75, the SD of difference = 1.22, and the dropout rate was 30%. As to feasibility of future studies, 110 patients were screened, but the actual number of entries was relatively small (n = 15); therefore, we think the inclusion criteria need to be relaxed to allow more patients to participate. Furthermore, in order to avoid a temporary decrease in hemoglobin after switching to an HIF-PHI, we recognized a necessity of starting treatment from a one-step-higher dose and advancing the timing of the dose increase of daprodustat. Although the formal components or methodological definition of a pilot study have not been established [38,39], this was not only a small-sample-size and exploratory study, but also included an estimate of the sample size, feasibility assessment, and consideration for the possibility for protocol modifications to achieve improved quality and efficiency in future large-scale study, so we think this clinical investigation was useful for this pilot study.

Our study has a several methodological limitations. First, the small sample size was a main limiting factor in this study and may have contributed to our failure to reach statistical significance for several of the metabolites (e.g., inositol and glucuronate) when comparing values before and after the switch to daprodustat. Furthermore, this work was a single-center study, lacked a control group, enrolled only Japanese subjects, and employed a medication (daprodustat) that has been clinically available for only a short time. Further trials will be needed to fully assess the effects of shifting patients from an ESA to an HIF-PHI, including the possible occurrence of long-term adverse events.

In the present study with hemodialysis patients, a comprehensive and non-targeted metabolomic analysis demonstrated that switching from an ESA to an HIF-PHI is associated with changes in phosphatidylinositol signaling, ascorbate and aldarate metabolism, and inositol phosphate metabolism. Previous studies have shown that HIF-PHIs can be administered orally and have an anemia-improving effect that is non-inferior to ESA. In addition, studies have shown that HIF-PHIs as improves iron and lipid metabolism and has the ability to raise hemoglobin levels in patients with chronic inflammation [40]. And, based on our findings, we further need to pay attention to changes in insulin resistance, glucose metabolism, skeletal muscle metabolism, and renal function when we start HIF-PHIs. Also, we need to conduct long-term and large-scale clinical trials to verify whether these effects are clinically significance.

## 4. Materials and Methods

### 4.1. Study Design

This exploratory pilot study was a single-center interventional study conducted at St. Luke’s International Hospital in 10 patients with CKD and anemia who were maintained on HD. Pleiotropic effects other than the erythropoietic effect (e.g., effects on glucose and lipid metabolism, and on inflammation status) were investigated comprehensively following a switch from an ESA to an HIF-PHI (daprodustat). The treatment period on daprodustat was set at three months, and non-targeted and comprehensive metabolomic analysis was performed using blood samples collected after the completion of the study protocol.

The study objective and primary endpoint was to compare the effects of a change from darbepoetin to daprodustat on the metabolomic profiles of HD-dependent patients during a 3-month period. Secondary endpoints included the following: mean change in hemoglobin from baseline; average monthly IV iron dose per participant: safety and tolerability (including incidence and severity of adverse events); reasons for the discontinuation of daprodustat; changes in blood pressure and heart rate; changes in iron parameters such as ferritin, TSAT, and IV iron dose; and changes in lipid and glucose parameters (triglycerides, LDL cholesterol, and plasma glucose).

Eligible participants were adults with anemia of CKD who were being maintained on HD and were receiving an ESA. Specifically, the inclusion criteria were as follows: (1) male or female, minimum 18 years of age; (2) not pregnant and not breastfeeding; (3) use of an ESA for at least 8 weeks prior to screening and continuing until trial initiation; (4) receiving three-times-weekly outpatient HD for at least 90 days; (5) hemoglobin at screening in the range of 10.0–12.0 g/dL; and (6) capable of giving signed informed consent. Exclusion criteria included the following: (1) ferritin < 100 ng/mL or TSAT < 20%; (2) anemia other than anemia of CKD (e.g., bone marrow aplasia or gastrointestinal bleeding); (3) active infection; (4) history of malignancy within the preceding 2 years; (5) myocardial infarction, acute coronary syndrome, stroke, or transient ischemic attack within 8 weeks prior to screening; (6) chronic heart failure with reduced ejection fraction (EF) or uncontrolled hypertension; (7) chronic liver disease, or hepatic or biliary abnormalities; or (8) PD + HD combination therapy. Stopping criteria included the following: (1) liver chemistry abnormalities; (2) diagnosis of cancer; (3) kidney transplant; (4) switch to PD or PD + HD combination therapy or home hemodialysis; or (5) pregnancy.

Blood samples were collected at Day 1 (immediately before daprodustat induction) and every 4 weeks thereafter through Week 12 (3 months). Blood also was collected from each subject just before, and 3 months after, the start of daprodustat administration and evaluated for the following parameters: hemoglobin, ferritin, serum iron, TIBC, TSAT, serum albumin, plasma glucose, LDL cholesterol, triglyceride, and CRP. For metabolomic analysis, additional samples were collected just before starting, and following completion of, the protocol. The samples collected at the first and second months were used only to confirmed stabilization of hemoglobin level and safety profile based on results from RCTs of HIF-PHIs versus ESAs in hemodialysis patients [4,41].

The scheme of daprodustat and darbepoetin of this study was based on results of a randomized, open-label phase III study which evaluated the efficacy of daprodustat switching from ESA [42]. Basically, darbepoetin 40 microgram per week was replaced with daprodustat 4 mg per day, and a dosage was adjusted based on an individual judgment by the attending physician based on the results of monthly blood tests (Figure 5). However, two patients with lower hemoglobin levels than the guideline recommended level were started with daprodustat 6 mg per day, and one with higher hemoglobin levels was started with 2 mg per day. At the end of the second month and third month, 4.2 ± 2.3 mg and 5.2 ± 3.7 mg of daprodustat were prescribed, respectively.

The median dose of darbepoetin alpha before the study initiation was 40 micrograms per week. Switching to daprodustat was started at 4 mg per day and then gradually increased according to the judgment by attending physician.

This study was conducted in accordance with the Declaration of Helsinki and its subsequent amendments, and according to a protocol approved by the Institutional Review Board of St. Luke’s International Hospital (Protocol Code: 21R-092; date of approval: 26 August 2021) and Tohoku University Hospital (Protocol Code: 2022-1-774; date of approval: 13 December 2022). Informed consent was obtained from all subjects involved in the study. Written, informed consent to publish this paper was obtained from the patients. The study was initiated on 25 November 2021 and completed on 12 April 2022.

### 4.2. Sample Preparation for GC-MS Measurements

The methods here reference previously reported methods [43,44,45]. An aliquot (50 µL) of plasma was mixed with 250 µL of a solution containing 55% methanol and 22% chloroform dissolved in distilled water containing 0.045 mg/mL 2-isopropylmalate (the internal standard). Notably, 2-Isopropylmalate is an intermediate formed during leucine biosynthesis, and it can be employed as an internal standard in GC-MS-based human serum metabolomics [46]. The resulting mixtures were incubated in a Thermomixer C (Eppendorf, Hamburg, Germany) for 30 min at 37 °C with shaking (1200 rpm). The samples then were centrifuged (3 min, 4 °C, 16,000× *g*), and 200 µL of the resulting supernatant was combined with an equal volume of distilled water. These mixtures were centrifuged again (3 min, 4 °C, 16,000× *g*), and 250 μL of the resulting supernatant was lyophilized to dryness under reduced pressure.

Next, the samples were oximated by resuspending the lyophilizate in 80 μL of methoxyamine hydrochloride (Sigma-Aldrich, Tokyo, Japan) dissolved at 20 mg/mL in pyridine, sonicating for 20 min, and shaking for 90 min at 1200 rpm and 30 °C. Methoxyamine hydrochloride is a reagent for preparation of O-methyl oximes [47]. The samples then were derivatized by adding 40 μL of N-methyl-N-trimethylsilyl-trifluoroacetamide (MSTFA; GL Sciences, Tokyo, Japan), mixing for 30 min at 1200 rpm and 37 °C, and centrifuging (3 min, 4 °C, 16,000× *g*). MSTFA is used as a derivatization reagent for GC-MS analysis because it reacts with all protic functional groups to silylate, and its by-product is highly volatile [48]. An aliquot (1 μL) of the resulting supernatant was then subjected to GC-MS.

### 4.3. GC-MS/MS Measurements

These measurements were performed with reference to the previously reported methods [43,44,45]. GC-MS analysis was performed using a GC-MS QP2010 Ultra (Shimadzu, Kyoto, Japan) with a fused silica capillary column (BPX-5; 30 m × 0.25 mm inner diameter, film thickness: 0.25 μm; Shimadzu), a front inlet temperature of 250 °C, and a helium gas flow rate through the column of 39.0 cm/s. The column temperature was maintained at 60 °C for 2 min, then raised by increments of 15 °C/min to 330 °C, and maintained at that temperature for 3 min. The interface and ion source temperatures were 280 °C and 200 °C, respectively. All data obtained by GC-MS analysis were analyzed using MetaboAnalyst software (v. 5.0; Reifycs, Inc., Tokyo, Japan). The retention times indicated in the Smart Metabolites Database (Shimadzu) were used as references to create a library for data analysis. To perform a semi-quantitative assessment, the peak area of each quantified compound was calculated and normalized using the 2-isopropylmalate peak area.

### 4.4. Statistical Analysis

Measurement results are presented as numbers (percentages) for categorical variables, as the mean ± standard deviation (SD) for normally distributed variables, and as the median with interquartile range (IQR) for non-normally distributed variables, unless indicated otherwise. IBM SPSS Statistics software (v. 29.0; IBM, Armonk, NY, USA) was used for data analysis, including paired *t*-tests, Mann–Whitney tests, and chi-squared tests. All tests were two-tailed. Values of *p* < 0.05 were considered statistically significant.

## 5. Conclusions

In summary, this study provided an integrative analysis of metabolomic data using blood samples from maintenance HD patients and detected characteristic changes in metabolite levels associated with a switch from an ESA (darbepoetin alfa) to an HIF-PHI (daprodustat). Our results provide detailed insights into the systemic biological effects of HIF-PHI dosing in HD patients, and these data may facilitate our understanding of the potential side effects that can be expected from the long-term use of this type of medication. Long-term and large-scale clinical trials to elucidate the pleiotropic effects of HIF-PHIs will be needed in the future.

## Figures and Tables

**Figure 1 ijms-24-12752-f001:**
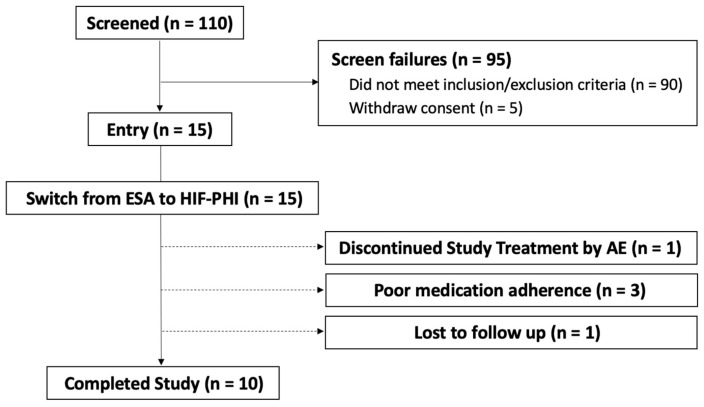
Participant disposition.

**Figure 2 ijms-24-12752-f002:**
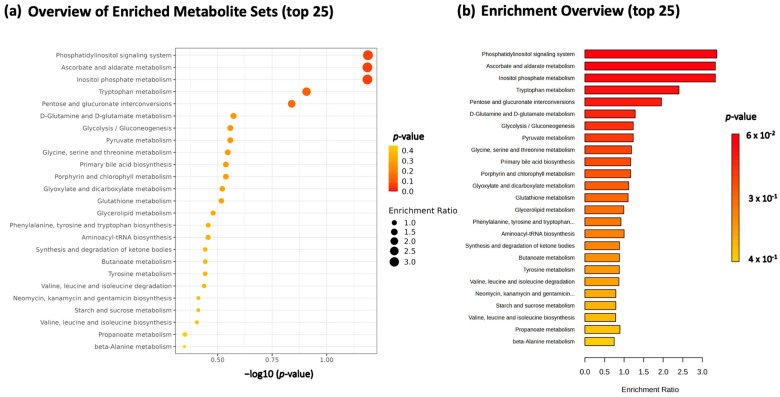
Summary of findings of changes in metabolomic profiles in HD patients following the switch to a hypoxia-inducible factor prolyl hydroxylase inhibitor. This figure shows changes detected by the comprehensive investigation of metabolomic profiles following the switch from an erythropoiesis-stimulating agent (darbepoetin alfa) to a hypoxia-inducible factor prolyl hydroxylase inhibitor (daprodustat). Significant changes in metabolites of phosphatidylinositol signaling, ascorbate and aldarate metabolism, and inositol phosphate metabolism were detected by this analysis. (**a**) Overview of enriched metabolites sets. Each pathway is shown in order of *p* value. (**b**) Enrichment overview. Each pathway is shown in order of enrichment ratio. Abbreviations: HD, hemodialysis.

**Figure 3 ijms-24-12752-f003:**
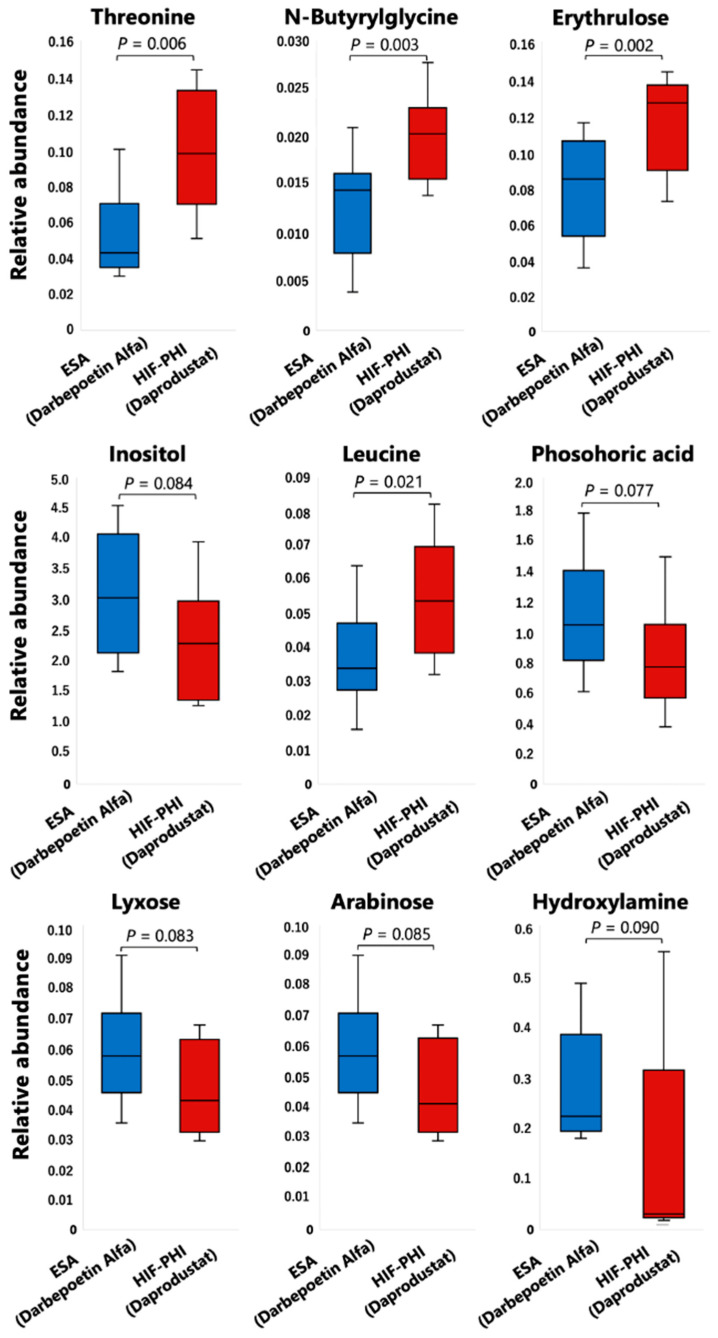
Nine metabolites whose levels were notably altered by a shift from an erythropoiesis-stimulating agent (darbepoetin alfa) to a hypoxia-inducible factor prolyl hydroxylase inhibitor (daprodustat).

**Figure 4 ijms-24-12752-f004:**
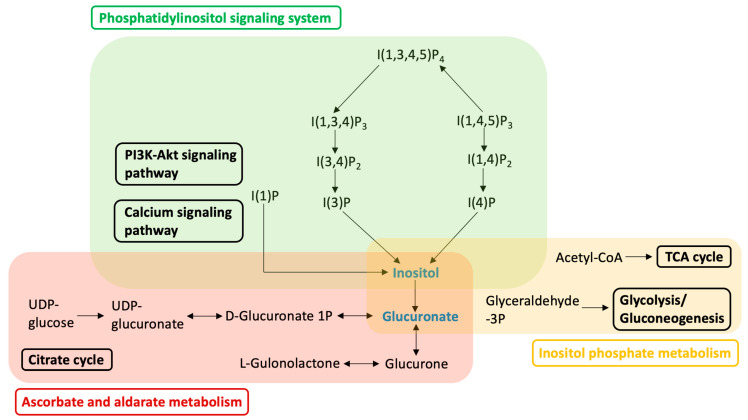
Schematic diagram of the pathways altered by a shift from an erythropoiesis-stimulating agent (darbepoetin alfa) to a hypoxia-inducible factor prolyl hydroxylase inhibitor (daprodustat).

**Figure 5 ijms-24-12752-f005:**
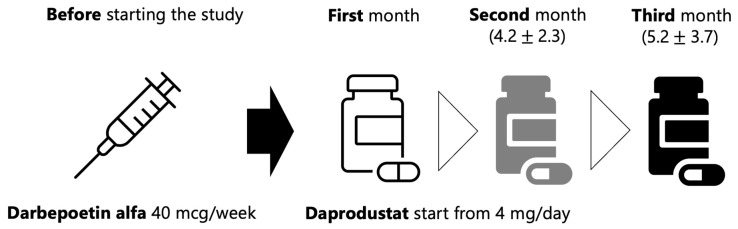
The scheme of daprodustat and darbepoetin of this study.

**Table 1 ijms-24-12752-t001:** Baseline characteristics of the 10 HD participants who completed the trial.

Characteristic	*n* = 10
Age, year	65.6 (11.3)
Men, *n* (%)	9 (90)
Race, *n* (%)	
Asian	10 (100)
Dialysis modality, *n* (%)	
Hemodialysis	10 (100)
Hemodiafiltration/hemofiltration	0 (0)
Dialysis vintage, year, *n* (%)	5.1 (4.4)
Postdialysis weight, kg	69.4 (15.4)
Postdialysis BP, mmHg	
Systolic BP	157.6 (20.3)
Diastolic BP	79.9 (11.4)
Prior ESA dose, darbepoetin alfa, mcg/w	41.0 (26.8)
Iron use, *n* (%)	
Intravenous iron	6 (60.0)
Oral only	4 (40.0)
No iron therapy	0 (0.0)
Hemoglobin, g/dL	11.1 (1.2)
Ferritin, ng/mL	117.1 (117.6)
Serum iron, mcg/dL	73.5 (24.1)
TIBC, mcg/dL	270.4 (42.0)
Transferrin saturation, %	28.5 (10.8)
Serum albumin, g/dL	3.7 (0.2)
Plasma glucose, mg/dL	116.4 (23.2)
LDL cholesterol, mg/dL	73.1 (15.0)
Triglyceride, mg/dL	80.3 (33.0)
CRP, mg/dL	0.13 (0.17)
History of cardiovascular disease, *n* (%)	3 (30.0)
History of stroke, *n* (%)	1 (10.0)
History of myocardial infarction, *n* (%)	1 (10.0)
History of heart failure, *n* (%)	1 (10.0)
History of thromboembolic events, *n* (%)	0 (0.0)
Smoking history, *n* (%)	
Never smoked	7 (70.0)
Current smoker	3 (30.0)
Former smoker	0 (0.0)
Statin use at entry, *n* (%)	6 (60.0)
Aspirin use at entry, *n* (%)	3 (30.0)
Vitamin K antagonist use at entry, *n* (%)	0 (0.0)
History of diabetes, *n* (%)	6 (60.0)

All parameters are expressed as either the number (percentage) or mean (SD). Abbreviations; BP, Blood pressure; ESA, erythropoiesis stimulating agent; TIBC, total iron binding capacity; LDL cholesterol, low density lipoprotein cholesterol; CRP, C reactive protein.

**Table 2 ijms-24-12752-t002:** Characteristics of the 10 HD participants who completed the trial, comparing values before and 3 months after switching from an ESA to an HIF-PHI.

Characteristic	Before (*n* = 10)	After (*n* = 10)	*p* Value
Postdialysis weight, kg	69.4 (15.4)	68.0 (15.2)	0.168
Postdialysis BP, mmHg			
Systolic BP	157.6 (20.3)	143.0 (19.0)	0.086
Diastolic BP	79.9 (11.4)	81.1 (15.2)	0.680
Iron use, *n* (%)			
Intravenous iron	6 (60.0)	0 (0.0)	
Oral only	4 (40.0)	4 (40.0)	
No iron therapy	0 (0.0)	6 (60.0)	
Hemoglobin, g/dL	11.1 (1.2)	9.8 (1.3)	0.086
Ferritin, ng/mL	117.1 (117.6)	152.8 (111.2)	0.500
Serum iron, mcg/dL	73.5 (24.1)	85.4 (24.6)	0.110
TIBC, mcg/dL	270.4 (42.0)	267.5 (40.4)	0.856
Transferrin saturation, %	28.5 (10.8)	32.5 (11.3)	0.308
Serum albumin, g/dL	3.7 (0.2)	3.6 (0.2)	0.159
Plasma glucose, mg/dL	116.4 (23.2)	121.2 (34.3)	0.603
LDL cholesterol, mg/dL	73.1 (15.0)	69.5 (21.1)	0.649
Triglyceride, mg/dL	80.3 (33.0)	91.9 (53.3)	0.338
CRP, mg/dL	0.13 (0.17)	0.21 (0.29)	0.345

All parameters are expressed as either the number (percentage) or mean (SD). A paired *t*-test was conducted for the comparison of each parameter before and after switching from an ESA to an HIF-PHI. Abbreviations; ESA, erythropoiesis stimulating agent; HIF-PHI, hypoxia-inducible factor prolyl hydroxylase inhibitor; BP, Blood pressure; TIBC, total iron binding capacity; LDL cholesterol, low density lipoprotein cholesterol; CRP, C reactive protein.

## Data Availability

The data presented in this study are available as Appendix A.

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
