# Peer review of "Changes in Metabolomic Profiles Induced by Switching from an Erythropoiesis-Stimulating Agent to a Hypoxia-Inducible Factor Prolyl Hydroxylase Inhibitor in Hemodialysis Patients: A Pilot Study"

_ijms, 2023, doi:10.3390/ijms241612752_

Round 1

Reviewer 1 Report

It is striking that the phrase "a pilot study" is observed in the title. I ask authors to assess whether their report is a pilot study based on the following information:

·      Foster RL. What a pilot study is and what it is not. J Spec Pediatr Nurs 2013;18(1):1-2.

·      Lowe NK. What Is a Pilot Study? J Obstet Gynecol Neonatal Nurs 2019;48(2):117-118.

·      In J. Introduction of a pilot study. Korean J Anesthesiol 2017;70(6):601-605.

·      Lancaster GA, Dodd S, Williamson PR. Design and analysis of pilot studies: Recommendations for good practice. J Eval Clin Pract 2004;10(2):307-312.

·      Hopkins WG. Quantitative research design. Sportscience 2000;4(10).

This is important because pilot studies can play an important part in knowledge development but are also “misused, mistreated, and misrepresented” (Lancaster et al., 2004). Please discuss.

Regarding the topic addressed by authors this is interesting because currently, one of the main health issues is chronic kidney damage, since 850 million people suffer from it and it is estimated that in the coming years it will be the fifth leading cause of death worldwide (Copur S, et al. Novel strategies in nephrology: what to expect from the future? Clin Kidney J 2022;16(2):230-244).

Therefore, understanding the processes related to this damage as well as establishing preventive or therapies are very important. 

Since patients with chronic renal failure suffer from anemia, among other alterations, the treatment applied to them consists of erythropoiesis-stimulating agents (as darbepoetin) and iron supplementation, but the use of drugs as hypoxia-inducible factor prolyl hydroxylase inhibitors (as daprodustat) that promote both synthesis of erythropoietin and proteins related with iron metabolism seems adequate. 

There are many reports about the beneficial effects of daprodustat on chronic kidney disease, and some of them are the following:

·      Singh AK, et al.  Daprodustat for the treatment of anemia in patients undergoing dialysis. N Engl J Med 2021;385:2325-35.

·      Singh AK, et al.  Daprodustat for the treatment of anemia in patients not undergoing dialysis. N Engl J Med 2021;385:2313-24.

·      Akizawa T, et al.  Efficacy and safety of daprodustat compared with darbepoetin alfa in Japanese hemodialysis patients with anemia: a randomized, double-blind, phase 3 trial. Clin J Am Soc Nephrol 2020;15:1155-65.

·      Tsubakihara Y, et al.  A 24-Week anemia correction study of daprodustat in Japanese dialysis patients. Ther Apher Dial 2020;24:108-14.

However, as the authors indicate, the pleiotropic effects of hypoxia-inducible factor prolyl hydroxylase inhibitors have not been fully studied. 

Authors then analyzed the effect of switching from darbepoetin, an erythropoiesis-stimulating agent, to a hypoxia-inducible factor prolyl hydroxylase inhibitor, daprodustat, on the metabolomics, biochemical profile, and inflammatory and iron parameters in the blood from patients on hemodialysis to determinate the cellular pathways involved in the improvement of the disease.

Data obtained from 10 adult patients with chronic renal failure and anemia (selected from 110 screened patients), maintained on hemodialysis, administered with iron, treated at least for 8 weeks with darbepoetin (an erythropoiesis-stimulating agent) and after administered with daprodustat for 3 months are shown in two tables and three figures.

Metabolomic analysis showed many not significant changes in pathways related to phosphatidylinositol signaling, as well ascorbate, aldarate, inositol phosphate, tryptophan, pentose and glucuronate metabolism (showed in figure 2) but significant increases were founded in four metabolites: threonine, N-butyrylglycine, erythrulose, and leucine (showed in figure 3). 

After stating the limitations of their study, the authors mention that results provided an integrative analysis of metabolomic data which may facilitate the understanding of the potential side effects due to long-term use of this hypoxia-inducible factor prolyl hydroxylase inhibitors. Additionally, they highlight that further studies are required to explain the significance of these changes in patients with chronic kidney disease.

Minor comments.

Please specify the scheme of administration of daprodustat and darbepoetin.

It is indicated that blood samples were collected every 4 weeks and that the measurements were made in samples of 3 months after start of daprodustat treatment. Was it not considered to use the samples collected at the first and second months?

Reviewer 2 Report

The article in question presents a scientific study, and its methodology section is particularly detailed, describing the inclusion and exclusion criteria for subjects, the process of blood sample collection, and the GC-MS/MS measurements. While the methodology is comprehensive, some terms and procedures might be unclear to a non-specialized audience. Adding explanations or references for specific techniques or chemicals used could enhance clarity. Ensuring that all necessary details are included for another researcher to replicate the study is vital, and if any proprietary or specialized equipment is used, providing alternatives would be beneficial. The text also mentions ethical approvals and informed consent, but these should be detailed enough to meet standard ethical guidelines.

Moving to the results section, the text briefly mentions findings related to phosphatidylinositol signaling, ascorbate and aldarate metabolism, and inositol phosphate metabolism. Though the mention is brief, the results section could benefit from more detail. Expanding on the findings, including statistical significance, effect sizes, and comparisons to existing literature, would add depth. If not already included, visual aids like tables or figures could visually represent the key findings, enhancing understanding.

The discussion and conclusion sections are essential for synthesizing the research and drawing meaningful insights. While the text provided does not include detailed information about these sections, they should connect the findings to broader theories or frameworks. The implications of the research, both practical and theoretical, should be articulated, and any limitations acknowledged. Suggestions for future research and a concise conclusion encapsulating the key findings would add impact.

The scholarly rigor of a research article is often reflected in its citations and references. Ensuring that all statements, methods, and conclusions that require citation are properly referenced is vital. The reference list should be complete, consistently formatted, and adhere to the chosen citation style. Providing references for specific techniques or theories would enhance the scholarly value of the document.

Finally, the overall accessibility and presentation of the article should be considered. Technical terms and jargon should be explained or defined, especially if the article is intended for a broader audience. Consistency in terminology, units of measurement, and formatting should be maintained. Professional proofreading and editing could ensure grammatical accuracy and stylistic coherence. If supplementary materials are extensive, organizing them in an easily accessible manner or providing links to an online repository would enhance the reader's experience. The integration of these elements would contribute to a polished and reader-friendly article that effectively communicates the research to the intended audience.

Well written

Round 2

Reviewer 1 Report

The manuscript was clearly improved.